# Advancing Tool-Augmented Large Language Models: Integrating Insights from Errors in Inference Trees

**Sijia Chen**[1,2,*]   **Yibo Wang**[1,2,*]   **Yi-Feng Wu**[3]   **Qing-Guo Chen**[3]
**Zhao Xu**[3]   **Weihua Luo**[3]   **Kaifu Zhang**[3]   **Lijun Zhang**[1,4,2,†]

[1]National Key Laboratory for Novel Software Technology, Nanjing University, Nanjing, China
[2]School of Artificial Intelligence, Nanjing University, Nanjing, China
[3]Alibaba International Digital Commerce   [4]Pazhou Laboratory (Huangpu), Guangzhou, China
{chensj, wangyb, zhanglj}@lamda.nju.edu.cn
{yixin.wyf, qingguo.cqg, changgong.xz,
weihua.luowh, kaifu.zkf}@alibaba-inc.com

## Abstract

Tool-augmented large language models (LLMs) leverage tools, often in the form of APIs, to improve their reasoning capabilities on complex tasks. This enables them to act as intelligent agents interacting with the real world. The recently introduced ToolLLaMA model by Qin et al. [2023] utilizes the depth-first search-based decision tree (DFSDT) mechanism for multi-step reasoning with 16000+ real-world APIs, effectively enhancing the performance of tool-augmented LLMs compared to traditional chain reasoning mechanisms. However, their approach only employs successful paths from decision trees (also called inference trees) for supervised fine-tuning (SFT), missing out on the potential learning opportunities from failed paths. Inspired by this, we propose an inference trajectory optimization framework based on preference learning to address this limitation. We first introduce a novel method for constructing step-wise preference data from tree-like expert trajectories, which leverages the previously ignored failed explorations in the decision trees. In the subsequent training phase, we first fine-tune the LLM with successful tool-usage expert trajectories and then apply direct preference optimization (DPO) with the preference data to update the LLM's policy, resulting in our ToolPrefer-LLaMA (TP-LLaMA) model. This approach not only enhances the utilization of original expert data but also broadens the learning space of the model. Our experiments demonstrate that by obtaining insights from errors in inference trees, TP-LLaMA significantly outperforms the baselines across almost all test scenarios by a large margin and exhibits better generalization capabilities with unseen APIs. At the same time, TP-LLaMA has also demonstrated superior reasoning efficiency compared to the baselines, making it more suitable for complex tool-usage reasoning tasks.

## 1 Introduction

In recent years, large language models (LLMs) have exhibited impressive capabilities in various areas, including language understanding and generation, multi-modal content learning and reasoning, and even embodied intelligence task processing [Brown et al., 2020, Alayrac et al., 2022, Zeng et al., 2023, Li et al., 2023, Chen et al., 2024, Liu et al., 2024a, Lu et al., 2024, Zhang et al., 2024, Cao et al., 2024a,b, Mazzaglia et al., 2024]. Despite these notable strengths, these models still

---

*Equal contribution. Work done during the internship at Alibaba International Digital Commerce.
†Corresponding author.

38th Conference on Neural Information Processing Systems (NeurIPS 2024).

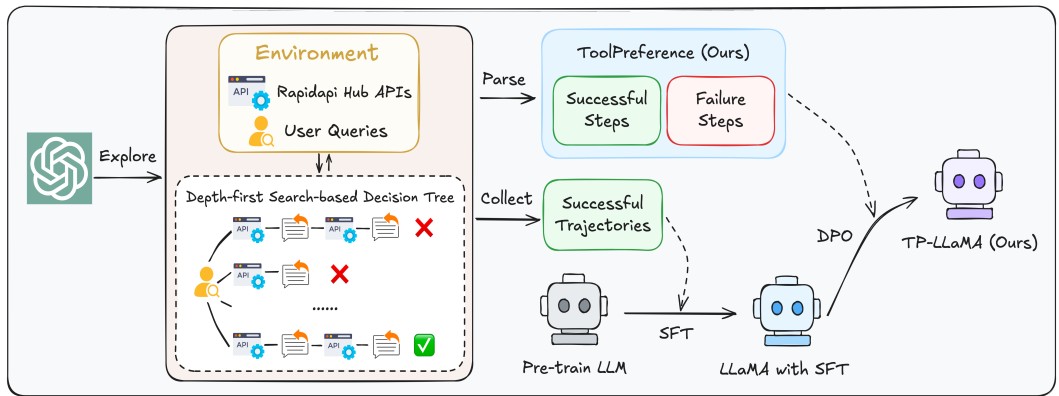

Figure 1: Our Inference Trajectory Optimization Framework.

face significant challenges, such as a lack of access to real-time information [Komeili et al., 2021] and difficulties in precise mathematical tasks [Patel et al., 2021, Lu et al., 2023b]. The development of tool-augmented LLMs tackles these challenges by enabling LLMs to interact with external tools (often in the form of APIs), significantly enhancing their capabilities. This advancement allows LLMs to serve as efficient intermediaries between users and a large ecosystem of applications. Notably, tool-augmented LLMs based on the ChatGPT [Brown et al., 2020] and GPT-4 [Achiam et al., 2023] have achieved outstanding results by using few-shot or zero-shot prompts to activate the LLM's inherent tool-usage abilities [Deng et al., 2023, Lu et al., 2023a, Lin et al., 2024]. Despite this progress, some studies demonstrate that open-source LLMs still exhibit a significant gap in their capacity to utilize external tools compared to state-of-the-art (SOTA) closed-source models like GPT-4 [Liu et al., 2024b, Wang et al., 2024b]. To bridge this gap, aligning these open-source LLMs with tool-usage downstream tasks is essential.

Currently, most efforts to align open-source LLMs with tool-usage downstream tasks rely on supervised fine-tuning (SFT) with expert trajectory datasets, which trains LLMs to learn strategies for subsequent actions based on previous actions and observations [Patil et al., 2023, Schick et al., 2023]. Early studies in this field typically have limitations such as a restricted variety of APIs, the reliance on single-tool scenarios, and the use of simple reasoning methods [Wei et al., 2022, Yao et al., 2023, Patil et al., 2023]. The recent work by Qin et al. [2023], which focuses on the scene of LLM's multi-step reasoning with external tools, solves the above limitations. They introduce an instruction tuning dataset called ToolBench, which includes over $16,000$ real-world APIs and various realistic instructions, along with expert trajectories annotated by ChatGPT based on a depth-first search-based decision tree (DFSDT) reasoning mechanism. They then perform SFT training on LLaMA with this dataset to create the ToolLLaMA model, which shows remarkable performance. However, ToolLLaMA's training is still based on expert behavior cloning, potentially limiting exploration of the target space and leading to suboptimal strategies. Additionally, although their expert trajectories are structured as DFS trees, only successful trajectories are utilized in the SFT training, which neglects valuable insights from failed attempts and results in low data utilization.

As the saying goes, "a fall into a pit, a gain in your wit", effective human learning involves not only drawing lessons from success but also from failures. Inspired by this, we propose a new inference trajectory optimization framework for developing tool-augmented LLMs as illustrated in Figure 1, which enhances the tool learning process by incorporating previously ignored failure exploration information via preference learning. Specifically, using the tree-like expert trajectories from ToolBench [Qin et al., 2023], we first parse each pair of branch nodes along the successful trajectory in the decision tree into a preference sample pair, thereby constructing a step-wise tool-usage preference dataset named *ToolPreference*.[3] Subsequently, after conducting SFT training on the pre-trained LLM with successful trajectories, we employ the direct preference optimization (DPO) method [Rafailov et al., 2023] with the ToolPreference dataset to further align the LLM with tool-usage downstream tasks, and thus obtain our model, named *ToolPrefer-LLaMA (TP-LLaMA)*. Our strategy improves the utilization of expert data and simultaneously broadens the learning space.

---

[3]The dataset is available at https://huggingface.co/datasets/chrissiecsj/ToolPreference.

Our experiments are conducted on the test tasks from ToolBench. To evaluate the performance, we adopt two metrics: the *pass rate*, which measures the probability of the model successfully providing an answer within limited steps; and the *win rate*, which quantifies the likelihood that the evaluator will prefer the model's responses. From the experiment results, we have the following findings:

- Across all test scenarios, TP-LLaMA consistently surpasses ToolLLaMA and other baselines, with an average pass rate improvement of at least 12% and a win rate that outperforms nearly all other models by an average of 4%. These results demonstrate that learning from failed attempts can significantly enhance the decision-making ability of LLMs. Additionally, our model shows superior generalization to unseen APIs.

- Efficiency experiments show that our model requires an average of only 22.62 steps for DFSDT inference, compared to 32.06 steps for the SFT model. This enhancement stems from our method's ability to avoid unnecessary branch explorations in DFSDT reasoning.

- Our ablation experiments verify that the effectiveness of our preference dataset and inference trajectory optimization framework has nothing to do with the base model itself. Better results can still be obtained after replacing the base model with Mistral-7B [Jiang et al., 2023], Qwen1.5-7B [Bai et al., 2023], and Gemma-7B [Team et al., 2024].

In summary, this work aims to enhance the performance of LLMs on multi-step reasoning with external tools by integrating insights from errors in tree-like reasoning trajectories and employing step-wise preference pairs for preference learning. Our key contributions include: (i) A novel method for constructing step-wise preference data from tree-like expert trajectories, which may provide inspiration for future research; (ii) The proposal of using the preference learning method DPO to optimize the LLM's tool-usage ability, along with the development of the TP-LLaMA model; (iii) Extensive experimental evaluations and in-depth analyses of the TP-LLaMA model, providing evidence of its effectiveness and validating its performance across various dimensions.

## 2 Related work

In this section, we briefly review recent progress on tool-augmented large language models and the development of preference learning.

**Tool-augmented large language models.** Over the past year, extensive research has been dedicated to developing tool-augmented LLMs, which exhibit improved reasoning abilities across various tasks by integrating external tools [Patil et al., 2023, Lu et al., 2023a, Schick et al., 2023, Lin et al., 2024]. The workflow for tool-augmented LLMs typically involves four key stages: task planning, tool selection, tool calls, and response generation. Early research mainly uses few-shot or zero-shot prompting methods to activate LLM's inherent tool-usage abilities, often employing GPT as the LLM agent to manage several external tools such as AI models, web search, Python, and more [Shen et al., 2023b, Lu et al., 2023a]. While GPT performs well with external tools, open-source LLMs like LLaMA often struggle with direct tool usage and need additional task alignment. Therefore, subsequent research often utilizes instruction-tuning datasets annotated with tool calls to train open-source models, enhancing their ability to use tools. At the same time, these studies continue to explore a wider range of tools and scenarios [Schick et al., 2023, Patil et al., 2023].

One of the most comprehensive efforts in this field is by Qin et al. [2023]. They initially collect $16,464$ real-world APIs across $49$ categories, then utilize ChatGPT to automatically generate instructions that could invoke these APIs, and annotate expert trajectories to create a high-quality instruction tuning dataset named ToolBench. During the annotation, they employ the DFSDT reasoning mechanism to broaden the search space and enhance reasoning capabilities. By fine-tuning LLaMA on ToolBench, they develop ToolLLaMA, which has shown a compelling capability to handle both single-tool and complex multi-tool instructions.

**Preference learning** Preference learning uses human preferences from feedback data to assist decision-making. The earliest research in this field employs specially designed neural networks to help agents optimize action choices based on structured human guidance in programming languages [Maclin and Shavlik, 1996]. Subsequent studies shift focus to learning from numerical rewards provided by humans and performing reinforcement learning based on the prediction of these

rewards [Isbell et al., 2006, Knox and Stone, 2008, Knox, 2012]. This approach finds applications in areas like embodied intelligence [Pilarski et al., 2011, Suay and Chernova, 2011] and dialogue systems [El Asri et al., 2016]. The introduction of preference-based reinforcement learning marks a key milestone in the field, which uses qualitative human preferences, often in the form of rankings, to guide the optimization of policy models [Akrour et al., 2011, Cheng et al., 2011]. Following this idea, Christiano et al. [2017] propose reinforcement learning from human feedback (RLHF), where a reward model is derived from human preferences to enhance reinforcement learning. This technique is later extended to natural language generation tasks [Kreutzer et al., 2018, Ziegler et al., 2019], advancing the integration of preference learning with LLM research [Ouyang et al., 2022].

## 3 Preliminaries

In this section, we start by formally defining the problem setup, and then we introduce key knowledge about preference learning methods, which is relevant to our approach.

### 3.1 Problem setup

In this work, we use an iterative paradigm for the LLM's multi-step reasoning with external tools, where the model selects each tool call based on the previous response, rather than pre-planning all tool calls at the start. Formally, we define it as a state transition process. The environment consists of a set of available tools $\mathcal{T} = \{T_1, T_2, \ldots, T_n\}$, each with specific functionalities accessible through API calls. The task begins with an initial instruction $I$, usually consisting of a user query and a system prompt. At each reasoning step $t$, the LLM processes the current context $S_t$, defined as:

$$S_t = \{I, H_t\}$$

where $H_t$ is the previous history, which includes the API decisions made $\{A_1, \cdots, A_{t-1}\}$, and the API responses received $\{R_1, \cdots, R_{t-1}\}$:

$$H_t = \{(A_1, R_1), \ldots, (A_{t-1}, R_{t-1})\}.$$

The LLM then generates an action decision $A_t$ based on this context, specifying both the tool $T_i \in \mathcal{T}$ to use and its parameters. After the tool executes, the response $R_t$ is generated and used to update the context. The reasoning process continues until the LLM determines that the task is complete and produces a final output $O$ to answer the original query or gives up the task.

### 3.2 Direct Preference Optimization

Preference learning has gained growing attention in LLM research. Its main goal is to optimize model outputs based on human (or expert) preferences, better aligning the model's behavior with the expectations of real-world applications. Assume there is a preference dataset defined as $\mathcal{D} = \{(x^{(i)}, y_w^{(i)}, y_l^{(i)})\}_{i=1,\ldots,|\mathcal{D}|}$, where $x^{(i)}$ denotes the $i$-th prompt, $y_w^{(i)}$ and $y_l^{(i)}$ denote the corresponding preferred and dispreferred output respectively. Moreover, the notation $y_w \succ y_l \mid x$ indicates that $y_w$ is preferred than $y_l$ for prompt $x$. Because the true distribution of human preferences is inaccessible, we assume it is generated by a latent reward model $r^*(x, y)$, where higher rewards indicate stronger preferences. Then, according to Rafailov et al. [2023], the human preference distribution $p^*$ can be captured by the Bradley-Terry (BT) model [Bradley and Terry, 1952]:

$$p^*(y_1 \succ y_2 \mid x) = \frac{\exp(r^*(x, y_1))}{\exp(r^*(x, y_1)) + \exp(r^*(x, y_2))} = \sigma(r^*(x, y_1) - r^*(x, y_2)),$$

where $\sigma$ is the logistic function. Obviously, we can estimate the parameters of the reward model via maximum likelihood estimation (equivalent to minimizing the negative log-likelihood.):

$$\mathcal{L}_R(r_\phi, \mathcal{D}) = -\mathbb{E}_{(x, y_w, y_l) \sim \mathcal{D}}[\log\sigma(r_\phi(x, y_w) - r_\phi(x, y_l))], \quad (1)$$

where $r_\phi$ is a parameterized reward model.

To optimize the inference trajectories of LLM based on human preference, a popular method in recent LLM research is Reinforcement Learning from Human Feedback (RLHF) [Christiano et al., 2017, Ouyang et al., 2022]. In the RL phase of this method, the optimization goal is

$$\max_{\pi_\theta} \mathbb{E}_{x \sim \mathcal{D}, y \sim \pi_\theta(y|x)}[r_\phi(x, y)] - \beta\mathbb{D}_{\text{KL}}[\pi_\theta(y \mid x) \| \pi_{\text{ref}}(y \mid x)], \quad (2)$$

where $r_\phi$ is the reward model learned before, $\pi_\theta$ is the policy model we need to optimize, $\beta$ is a weighting parameter that controls the deviation from the base reference policy model $\pi_{\text{ref}}$ (i.e., the LLM after SFT training). In practice, $\pi_\theta$ is also initialized to the LLM after SFT. RLHF will use reinforcement learning methods (such as PPO [Schulman et al., 2017]) to optimize (2) and update the LLM's strategy, with $r_\phi(x, y)$ providing reward feedback. Additionally, some research in multi-step reasoning scenarios trains process reward models to evaluate each step instead of the entire output [Ma et al., 2023, Wang et al., 2024a]. However, RLHF incurs significant computational overhead, long training times, and potential instability [Shen et al., 2023a, Rafailov et al., 2023], making it less suitable for general tool-usage tasks.

Therefore, we choose a more convenient and faster approach that can also effectively align the model's preferences — Direct Preference Optimization (DPO) [Rafailov et al., 2023], which eliminates the need to learn the reward model and directly uses preference data to optimize the LLM. Specifically, the optimal solution of (2) can be written as

$$\pi_r\left(y \mid x\right) = \frac{1}{Z(x)}\pi_{\text{ref}}\left(y \mid x\right)\exp\left(\frac{1}{\beta}r(x, y)\right), \tag{3}$$

where $Z(x) = \sum_y \pi_{\text{ref}}\left(y \mid x\right)\exp\left(\frac{1}{\beta}r(x, y)\right)$ is the partition function [Rafailov et al., 2023]. We rearrange (3) to express $r(x, y)$ in terms of $\pi_r$ and $\pi_{\text{ref}}$:

$$r(x, y) = \beta\log\frac{\pi_r\left(y \mid x\right)}{\pi_{\text{ref}}\left(y \mid x\right)} + \beta\log Z(x). \tag{4}$$

Substituting (4) into (1), we can finally get the learning goal of DPO

$$\mathcal{L}_{\text{DPO}}(\pi_\theta, \pi_{\text{ref}}) = -\mathbb{E}_{(x, y_w, y_l)\sim\mathcal{D}}\left[\log\sigma\left(\beta\log\frac{\pi_\theta\left(y_w \mid x\right)}{\pi_{\text{ref}}\left(y_w \mid x\right)} - \beta\log\frac{\pi_\theta\left(y_l \mid x\right)}{\pi_{\text{ref}}\left(y_l \mid x\right)}\right)\right],$$

where $\pi_\theta$ is a parametrized policy that we need to optimize. As a result, the optimization objective of DPO avoids additional learning of the reward model and the RL process while maximizing the final reward, which is more suitable for our general tool-usage scenarios.

# 4 Our method

In this section, we introduce our inference trajectory optimization framework, beginning with an overview of the framework, followed by a description of the preference data construction process.

## 4.1 The framework

Our framework is composed of two key stages: dataset construction and training. In the dataset construction stage, we create a tool-usage preference dataset, named ToolPreference, which is derived from the tree-like expert trajectories in Toolbench [Qin et al., 2023]. The specific process for constructing this dataset will be detailed in section 4.2.

**Remark 1** *It is important to emphasize that our preference data construction approach is not limited to Toolbench and can be adapted to any tree-structured multi-step instruction-tuning dataset, offering flexibility for various applications.*

In the training stage, we first perform SFT training on a pre-trained LLM using a resampled version of the instruction-tuning data from Toolbench (refer to Remark 2 for the resampling process). SFT training has been commonly adopted in previous research to enhance tool-augmented LLMs. However, mere cloning expert behavior through SFT is insufficient, as this method fails to adequately explore the environment, and can result in suboptimal strategies. To address this, after the SFT training, we further perform DPO training on the model with the ToolPreference dataset. This additional preference learning enhances the models reasoning capabilities when interacting with external tools and aligns its decision-making preferences with human preferences.

## 4.2 Preference data construction

Before introducing our preference data construction method, we first describe the dataset structure and expert trajectory format used in ToolBench [Qin et al., 2023].

- **Dataset structure.** ToolBench consists of two main components: API information data and instruction tuning data. The API information data is sourced from RapidAPI Hub[4] and includes $3,451$ tools across $49$ categories, with a total of $16,464$ APIs (as each tool can have multiple APIs). Each API entry contains detailed information such as the name, description, HTTP method, URL, required and optional parameters, and executable code snippets for API calls. This comprehensive data enables LLMs to perform few-shot inference with effective API calls. The instruction-tuning data includes various single-tool or multi-tool instructions as well as corresponding annotated expert trajectories, generated in a self-instruction method by ChatGPT.

- **Expert trajectory format.** While traditional LLMs often use sequential reasoning methods like chain-of-thought (CoT) [Wei et al., 2022], which follow a single path to completion, ToolBench adopts a depth-first search (DFS) reasoning approach. As shown in the left half of Figure 2, expert trajectories in ToolBench are structured as decision trees with each tree node representing an LLM decision about an API call. Based on the tree structure, ToolBench implements DFS reasoning using two techniques. First, it defines two additional functions: one is "Finish with final answer", where the LLM concludes it has gathered enough API responses to provide a correct answer and terminate the reasoning process, and the other is "Finish with giving up", where the LLM feels unable to proceed with the task, abandons the current path and returns to a previous node. Second, diversity prompts are used to expand the search space. When expanding child nodes, the LLM will be prompted with information about previously explored child nodes of the same layer, and explicitly encouraged to generate different ones. Consequently, the LLM is allowed to either abandon the current path and restart from a previous step or proceed along a more promising path, exploring until an answer is reached or the node limit is reached.

We employ the second release of ToolBench[5], which includes over $120,000$ expert trajectories. Our approach is designed based on the motivation of improving data utilization. Although the tree-like expert trajectories in ToolBench extensively search the answer space, only successful paths are used in their training, neglecting valuable insights from failure paths. To address this, we extract preference decision pairs from each tree-like expert trajectory. After filtering out trajectories without failed exploration branches, we explore two different construction methods:

- **Path-wise** means using an entire success path and an entire failure path in the same decision tree to form a preference pair. As shown in the upper right part of Figure 2, $\langle 0, 9, 12, 13, 14, 15 \rangle$ is the success path of the decision tree, and $\langle 0, 1, 2 \rangle$, $\langle 0, 3, 4, 5, 6 \rangle$, $\langle 0, 3, 7, 8 \rangle$, $\langle 0, 9, 10, 11 \rangle$ are 4 failure paths, so their Cartesian product can constitute a path-wise preference dataset, where $\succ$ denotes the left part is preferred than the right part.

- **Step-wise** means using each branch node along the success path in the tree and its corresponding pair of child nodes (which must contain a child node on the success path) to construct a preference pair. As shown in the lower right part of Figure 2, $\langle 0, 9, 12, 13, 14, 15 \rangle$ is the success path of the decision tree, while $0$ and $9$ are nodes with branches along the success path. Therefore, $\langle 0, 9 \rangle \succ \langle 0, 1 \rangle$, $\langle 0, 9 \rangle \succ \langle 0, 3 \rangle$, and $\langle 0, 9, 12 \rangle \succ \langle 0, 9, 10 \rangle$ can respectively form a preference pair.

Although it is intuitive and common to use path-wise preference samples, this approach is not well-suited to our task scenario. Theoretically, it may limit the model to only differentiate between correct and incorrect final responses to specific instructions, resulting in poor generalization with unseen instructions or tools. From an engineering perspective, learning preferences for an entire path at once is inconsistent with the model's reasoning mechanism of inferring the next API call based on the response of the previous API execution each time, which makes it inherently unsuitable for the DFSDT reasoning mechanism.

In contrast, the step-wise design highlights the differences between each reasoning step, providing the model with more fine-grained process supervision. Theoretically, this method can better adjust the model's reasoning process and enhance its generalization performance. It is also a more suitable fit for implementation within the DFSDT reasoning framework. Consequently, we create $69,393$ pairs of preference samples from ToolBench in a step-wise manner. Each pair is formatted

---

[4] https://rapidapi.com/hub
[5] https://github.com/OpenBMB/ToolBench.git

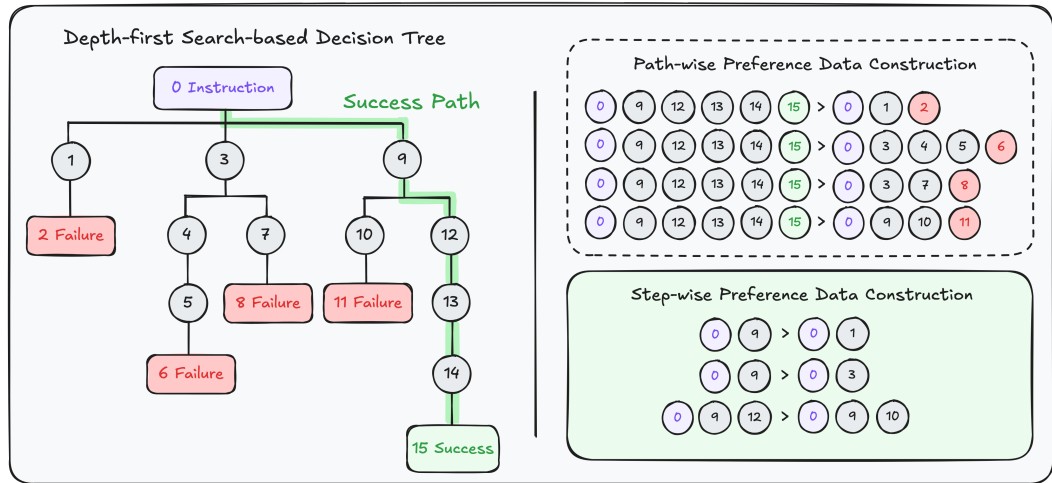

Figure 2: Depth-first search-based decision tree and two preference data construction methods

as {Instruction, Input, Output}. The Instruction includes the system prompt, detailing the DFSDT reasoning task and the relevant API documentation. The Input contains the user query and the reasoning history up to the current step, while the Output presents a preferred and a dispreferred reasoning step for the given input. Additionally, to prevent information leakage, we carefully remove any diversity prompts from each node's information during parsing.

**Remark 2** *To ensure a rigorous comparison between models with and without preference learning in subsequent experiments, we also do not directly use the instruction-tuning dataset provided by Toolbench during the SFT phase. Instead, we filter out expert trajectories lacking failed exploration branches, as these could not be parsed into preference samples, and resampled the remaining data to create our SFT training set. This ensures the training data distribution remains consistent across models, regardless of whether preference learning is applied.*

## 5 Experiments

In this section, we investigate the performance of our inference trajectory optimization framework. We first introduce the experiments settings in Section 5.1. We then present the main results in Section 5.2, the efficiency experiments in Section 5.3, and the ablation experiments in Section 5.4.

### 5.1 Experiments settings

**Evaluation metrics.** Since our model uses APIs from the online platform RapidAPI Hub, there may be changes such as version updates or service termination over time, making it difficult to provide a fixed solution path for each test instruction. Following Qin et al. [2023], we use *pass rate* and *win rate* as evaluation metrics in our experiments. The pass rate represents the proportion that the model successfully gives answers within a certain number of reasoning actions (set to 200 in our experiment).[6] Specifically, a sample is considered passed if the reasoning trajectory finishes with the "Finish with final answer" API call. Additionally, we filter out samples that yield meaningless answers using a predefined set of feature keywords, such as "sorry", "apologize", etc. The win rate measures the likelihood that the solution path provided by the test model is preferred over the reference solution path for the same instruction. We use the answers given by ChatGPT+DFSDT as the reference solution paths and employ ChatGPT to determine preference.[7]

---

[6]During our experiment process, we noticed that ToolBench has been updated with a revised definition of pass rate [Qin et al., 2024] The definition we use in the main text follows the original version, while the revised definition and corresponding results will be provided in Appendix B.1

[7]The ChatGPT version we used in the experiments in the main text is gpt-3.5-turbo-16k.

Table 1: Main Experiment Results. Avg represents the average pass rate or win rate of the 6 test scenarios. A win rate higher than 50% means the model performs better than ChatGPT+DFSDT.

| Pass Rate | | | | | | | |
|---|---|---|---|---|---|---|---|
| **Model** | **G1-Ins.** | **G1-Tool** | **G1-Cat.** | **G2-Ins.** | **G2-Cat.** | **G3-Ins.** | **Avg** |
| ChatGPT | 0.52 | 0.55 | 0.60 | 0.51 | 0.51 | 0.21 | 0.48 |
| Davinci | 0.49 | 0.47 | 0.45 | 0.40 | 0.27 | 0.29 | 0.40 |
| ToolLLaMA | 0.54 | 0.60 | 0.62 | 0.47 | 0.54 | 0.17 | 0.49 |
| LLaMA with SFT | 0.47 | 0.53 | 0.72 | 0.48 | 0.63 | 0.35 | 0.53 |
| **TP-LLaMA (ours)** | **0.55** | **0.65** | **0.80** | **0.62** | **0.67** | **0.61** | **0.65** |
| Win Rate | | | | | | | |
| **Model** | **G1-Ins.** | **G1-Tool** | **G1-Cat.** | **G2-Ins.** | **G2-Cat.** | **G3-Ins.** | **Avg** |
| ChatGPT | - | - | - | - | - | - | - |
| Davinci | 0.37 | 0.37 | 0.35 | 0.35 | 0.29 | 0.54 | 0.38 |
| ToolLLaMA | 0.55 | 0.53 | **0.57** | 0.56 | 0.52 | 0.68 | 0.57 |
| LLaMA with SFT | 0.54 | 0.51 | 0.56 | 0.65 | 0.57 | 0.81 | 0.61 |
| **TP-LLaMA (ours)** | **0.56** | **0.59** | 0.54 | **0.70** | **0.64** | **0.86** | **0.65** |

**Training settings.** For the 2-epoch SFT training, we randomly sampled $11,142$ instances from the expert-annotated data in ToolBench after removing those without failed exploration branches. The batch size is 16 and the learning rate is 1e-5 during SFT training. For the 1-epoch DPO training, we randomly sample $8,202$ preference data pairs from our ToolPreference dataset, the batch size is 8, the learning rate is 1e-6 and $\beta = 0.5$ in (2). It is important to note that our sampling is performed at the instruction level, which means that samples corresponding to the same instruction are either all included in the training set or none are included. We provide a detailed explanation of our design choices for training hyperparameters in Appendix A.1. All our experiments are conducted on a single machine equipped with 8 NVIDIA A100 GPUs with 80G memory.

**Testing settings.** We investigate six test scenarios same as Qin et al. [2023]: G1-Cat., G1-Ins., G1-Tool, G2-Cat., G2-Ins., and G3-Ins.. The specific meanings are as follows: (1) **G1**: instructions that only use a single tool; (2) **G2**: instructions that use intra-category multi-tools; (3) **G3**: instructions that use inter-category multi-tools; (4) **Cat. (Category)**: unseen tools that belong to the unseen category of tools in the training data; (5) **Ins. (Instruction)**: unseen instructions for the same set of tools in the training data; (6) **Tool**: unseen tools that belong to the same category of tools in the training data. Each test scene contains 200 test samples, except G3-Ins., which contains 100 test samples. The six test scenarios have different task difficulties and generalization challenges, which can well reflect the comprehensive performance of models.

**Baselines.** We compare our model with several models without preference learning. Among them, we select the expert model ChatGPT and OpenAI Text-Davinci-003 (Davinci for short) as baselines. In addition, we also show the results of ToolLLaMA and the model trained by SFT using our resampled SFT training set (LLaMA with SFT for short) for comparison. Note that all models here are combined with DFSDT for inference. In addition, regarding the ToolLLaMA results, we directly use the reasoning answers of ToolLLaMA on test sets provided by ToolBench's GitHub repository to calculate pass rates and win rates.

## 5.2 Main results

We employ LLaMA-2-7B as the base model of our training framework and finally obtain our model, named ToolPrefer-LLaMA (TP-LLaMA). The context length of LLaMA-2-7B is extended to 8192 tokens to accommodate our tool-usage reasoning tasks. The main results are shown in Table 1. We have the following important observations:

Table 2: Efficiency Results of TP-LLaMA. Imp denotes the improvement of TP-LLaMA over LLaMA with SFT in terms of the average steps.

| Model | G1-Ins. | G1-Tool | G1-Cat. | G2-Ins. | G2-Cat. | G3-Ins. | Avg | Imp |
|---|---|---|---|---|---|---|---|---|
| LLaMA with SFT | 32.82 | 34.60 | 31.45 | 31.98 | 35.05 | 26.44 | 32.06 | - |
| **TP-LLaMA (ours)** | **24.54** | **24.19** | **23.85** | **23.98** | **23.53** | **15.61** | **22.62** | **29.44%** |

- TP-LLaMA significantly outperforms LLMs without preference learning in terms of pass rate, demonstrating the best performance across all six test scenarios, with an average improvement of of over 12% compared to models not optimized using preference data.

- Regarding win rate, TP-LLaMA also exhibits competitive performance, just 3% below ToolLLaMA in the G1-Cat. scenario, while achieving the best results in all other scenarios.

- Furthermore, TP-LLaMA shows strong performance in more challenging task scenarios such as G2-Cat., G2-Ins., and G3-Ins., maintaining effectiveness similar to that in simpler tasks. Notably, in the G3-Ins. scenario, TP-LLaMA's pass rate increased by over 26%, proving that our DPO training process using preference data significantly enhances the model's ability to handle complex multi-tool tasks.

Although we use the provided reasoning answers of ToolLLaMA from ToolBench's GitHub repository to calculate its rates, the results indeed differ from those reported in their paper [Qin et al., 2023]. This may be due to the reasoning answers version not matching the one used in their paper or differences in the evaluation environment settings. However, it's important to emphasize that our results remain valid and reliable. We apply consistent settings across all models, so their relative differences are meaningful. Overall, our experimental results indicate that through preference learning, TP-LLaMA can master various tool-usage instructions better and exhibits stronger generalization capabilities to unseen tools, categories, and instructions.

## 5.3 Efficiency Evaluation

We also evaluate the inference efficiency of TP-LLaMA on six test scenarios and employ the average number of DFSDT inference steps required for samples that ended with the Finish function as the metric. From Table 2, we can find that LLaMA with SFT requires an average of 32.06 steps for reasoning, while our TP-LLaMA only requires an average of 22.62 steps of reasoning in all test scenarios, with an improvement of 29.44%. These results clearly demonstrate that the inference efficiency of TP-LLaMA is remarkably superior to that of the model trained only with success trajectories. This advantage arises from our step-wise preference data, which allows the model to identify the most optimal decisions at each step of reasoning through DPO training. As a result, the model avoids the exploration of unnecessary sub-optimal branches in the decision tree, thereby increasing reasoning speed and efficiency.

## 5.4 Ablation experiments

In the ablation experiments, to verify the effectiveness of our framework, we further replace LLaMA-2-7B with other base models, including Mistral-7B, Qwen1.5-7B, and Gemma-7B. The results are shown in Table 3 and Table 4.

From Table 3, no matter which base model is used, training on preference data can always bring gains to the performance of the model, which verifies the model-independent effectiveness of our framework. Specifically, in terms of pass rates, models that have learned from expert errors improve by at least 8% on average compared to those that only receive training on success trajectory information. Similarly, in terms of win rates, models with insights from preference data generally outperform those without preference learning. Table 4 further confirms that our method significantly improves model inference efficiency by a large margin, up to an average of 33.35%.

Table 3: Ablation Performance Experiment Results. Avg represents the average pass rate or win rate of the 6 test scenarios. A win rate higher than $50\%$ means the model performs better than ChatGPT+DFSDT.

| Pass Rate | | | | | | | |
|---|---|---|---|---|---|---|---|
| **Model** | **G1-Ins.** | **G1-Tool** | **G1-Cat.** | **G2-Ins.** | **G2-Cat.** | **G3-Ins.** | **Avg** |
| Mistral with SFT | 0.70 | 0.43 | 0.42 | 0.53 | 0.46 | 0.27 | 0.47 |
| TP-LLaMA (Mistral) | **0.71** | **0.53** | **0.55** | **0.70** | **0.64** | **0.57** | **0.62** |
| Qwen with SFT | 0.69 | 0.51 | 0.51 | 0.66 | 0.55 | 0.49 | 0.57 |
| TP-LLaMA (Qwen) | **0.77** | **0.53** | **0.60** | **0.72** | **0.61** | **0.65** | **0.65** |
| Gemma with SFT | 0.67 | 0.44 | 0.49 | 0.47 | 0.44 | 0.29 | 0.47 |
| TP-LLaMA (Gemma) | **0.80** | **0.48** | **0.61** | **0.70** | **0.65** | **0.68** | **0.65** |
| Win Rate | | | | | | | |
| **Model** | **G1-Ins.** | **G1-Tool** | **G1-Cat.** | **G2-Ins.** | **G2-Cat.** | **G3-Ins.** | **Avg** |
| Mistral with SFT | 0.52 | 0.47 | 0.55 | 0.61 | 0.61 | 0.72 | 0.58 |
| TP-LLaMA (Mistral) | **0.53** | **0.50** | **0.57** | **0.62** | **0.64** | **0.74** | **0.60** |
| Qwen with SFT | 0.53 | 0.52 | 0.52 | 0.64 | 0.66 | 0.75 | 0.60 |
| TP-LLaMA (Qwen) | **0.54** | **0.54** | **0.58** | **0.66** | **0.67** | **0.81** | **0.63** |
| Gemma with SFT | 0.58 | 0.54 | 0.53 | 0.50 | 0.62 | 0.73 | 0.58 |
| TP-LLaMA (Gemma) | **0.61** | **0.57** | **0.58** | **0.65** | **0.67** | **0.75** | **0.64** |

Table 4: Ablation Efficiency Experiment Results. Imp denotes the improvement of TP-LLaMA over LLaMA with SFT in terms of the average steps.

| Model | Average Number of Steps in One Successful Path | | | | | | | |
|---|---|---|---|---|---|---|---|---|
| | **G1-Ins.** | **G1-Tool** | **G1-Cat.** | **G2-Ins.** | **G2-Cat.** | **G3-Ins.** | **Avg** | **Imp** |
| Mistral with SFT | 28.92 | 26.65 | 30.22 | 25.69 | 26.58 | 25.24 | 27.22 | - |
| **TP-LLaMA (Mistral)** | **25.30** | **25.01** | **23.36** | **23.51** | **20.74** | **16.42** | **22.39** | **17.74%** |
| Qwen with SFT | 35.74 | 34.66 | 36.85 | 32.74 | 36.18 | 37.93 | 35.68 | - |
| **TP-LLaMA (Qwen)** | **25.12** | **23.83** | **24.49** | **23.84** | **26.92** | **22.18** | **24.40** | **31.61%** |
| Gemma with SFT | 27.49 | 22.77 | 24.10 | 18.70 | 20.52 | 21.19 | 22.46 | - |
| **TP-LLaMA (Gemma)** | **17.15** | **13.88** | **15.91** | **13.63** | **13.30** | **15.94** | **14.97** | **33.35%** |

## 6    Conclusion and future work

In this work, we propose a novel inference trajectory optimization framework that leverages preference learning to enhance the performance of tool-augmented LLMs. We first built a step-wise tool-usage preference dataset, ToolPreference, using our proposed new preference data construction method to convert previously ignored failed explorations in tree-like expert trajectories into valuable training data. After initial SFT training on the LLM, we use ToolPreference for DPO training to further refine the LLM's strategy, resulting in our TP-LLaMA model. Our extensive comparative experiments prove that TP-LLaMA significantly outperforms the baseline models in nearly all test scenarios by learning from single-step errors in inference trees. TP-LLaMA also exhibits superior generalization capabilities and efficiency. Furthermore, ablation experiments confirm the model-independent effectiveness of our framework. In future work, we will try to explore tool-learning research with more complex, human-like reasoning mechanisms, and incorporate preference learning for further optimization. We also aim to extend our research to multimodal scenarios to evaluate the broader effectiveness of our approach.

## Acknowledgments and Disclosure of Funding

This work was partially supported by NSFC (U23A20382, 62122037), and the Collaborative Innovation Center of Novel Software Technology and Industrialization.

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

# Appendix

## A  Experimental details

In this section, we supplement some details of the experiments in the main text, including the training details in Appendix A.1, the API information format in Appendix A.2, and the ToolPreference sample example in Appendix A.3.

### A.1  Details for training

**Training hyperparameters**  We provide an explanation of our design choices regarding training hyperparameters, specifically the sizes of the training and test sets. We first filter 42,192 tree-like expert trajectories with branching nodes from Toolbench, which leads to 69,393 DPO samples and 184,816 SFT samples after processing (as each instruction may correspond to multiple samples). After allocating a small part as a validation set, we sample training sets of different sizes based on these samples. The sampling methods we tried include "by instruction" and "by sample". For sampling by instruction, the size of the SFT training set ranges from 2,500 to 10,000 queries, and the size of the DPO training set ranges from 5,000 to 32,192 queries, yielding nine combinations. For sampling by sample, the size of SFT varies from 10,000 to 183,561, and the size of DPO varies from 10,000 to 68,951, yielding seven combinations. We conduct small-scale tests based on these different training settings and find that increasing the size may lead to decreased model performance in scenarios with strong generalization, such as G3-ins. (e.g., with settings {SFT: 44,412, DPO: 41,226}, the pass rate drops to 0.36), possibly due to overfitting. Consequently, we select the set {SFT: 11,142, DPO: 8,202} in our final experiments.

**Computation time consumption**  With 8 NVIDIA A100 GPUs, our SFT training phase takes an average of 4.6 hours, and the DPO training phase takes an average of 3.2 hours. In the inference phase, each API call takes about 3.4 seconds, and each query takes about 48.7 seconds. Computation time varies due to task complexity, network conditions, and API service status.

### A.2  Details for API Information

Below we provide a detailed document of the API collected in ToolBench to help readers understand the format and content of API information.

**API Information Sample**

```json
{
  "name": "Get Character By ID",
  "url": "Get individual character by ID\n Options:\n\n-
              Limit \u2192 Limit amount of responses received
              \n- Step \u2192 Skip amount of characters",
  "method": "GET",
  "required_parameters": [
    {
      "name": "id",
      "type": "NUMBER",
      "description": "",
      "default": ""
    }
  ],
  "optional_parameters": [],
  "code": "import requests ......",
  "test_endpoint": {
    "err": "Please enter a valid number for the character id."
  }
}
```

### A.3 Details for ToolPreference

Here we show an example preference pair in ToolPreference. For the sake of brevity, we have omitted some less important information, including some rules in the instruction, API parameter information, and some response content.

---

**Preference Sample Pair**

**Instruction**

```
You are AutoGPT, you can use many tools(functions) to do
the following task......Specifically, you have access to
the following APIs:

{"name": "get_amazon_product_details_for_abiola_amazon_data
_scraper", "description": "This is the subfunction for tool
 'abiola_amazon_data_scraper', you can use this tool. The
description of this function is: This endpoint get the
general product description of any amazon product.", "
parameters": ......}

{"name": "get_amazon_product_reviews_for_abiola_amazon_data
_scraper", "description": "This is the subfunction for tool
 'abiola_amazon_data_scraper', you can use this tool. The
description of this function is: This endpoint gives you
the review of the product you enter.", "parameters":
......}

{"name": "get_amazon_product_offer_for_abiola_amazon_data
_scraper", "description": "This is the subfunction for tool
 'abiola_amazon_data_scraper', you can use this tool. The
description of this function is: This endpoint gets the
offer of any amazon product.", "parameters": ......}
......
```

**Input**

```
{"from": "user", "value": "Can you fetch the product
details, reviews, and offers for the product with ID
B08L5NP6NG? Also, provide me with the search results for
the product 'Apple MagSafe Charger'.Begin!"}

{"from": "assistant", "value": "Thought: Action:
get_amazon_product_details_for_abiola_amazon_data_scraper
Action Input: {'productid': 'B08L5NP6NG'}"}

{"from": "function", "value": "{'name': 'Apple MagSafe
Charger', 'product_information': {'brand': 'Visit the Apple
 Store', 'brand_url': 'https://www.amazon.com/stores/Apple/
page/77D9E1F7-0337-4282-9DB6-B6B8FB2DC98D?ref_=ast_bln', '
full_description': '', 'pricing': '$39.00', ......}"}

{"from": "assistant", "value": "Thought: Action:
get_amazon_product_reviews_for_abiola_amazon_data_scraper
Action Input: {'productid': 'B08L5NP6NG'}"}

{"from": "function", "value": "......"}
```

---

# B  Supplementary experiment results

In this section, we first supplement the experiments on another defined pass rate in Appendix B.1. Then we conduct a qualitative analysis of our method in Appendix B.2, giving a specific case study.

## B.1  Another definition of pass rate

In the second version of ToolBench [Qin et al., 2024], a new pass rate definition using GPT-assisted evaluation is introduced. For each query, it first checks if the "Finish with Final Answer" API is called; if not, it's considered a failure. If it is, GPT evaluates whether the answer resolves the query. If successful, it is marked as a pass. If not, GPT further assesses whether the query is solvable with the available APIs. If it isn't, it is still considered a pass; otherwise, it is marked as a failure. Due to the ToolBench API server being offline temporarily, we use our own RapidAPI accounts to access APIs for evaluation experiments with the new pass rate definition. To maintain consistency in API status, we utilize ToolLLaMA's open-source model[8] to perform reasoning on the test sets, instead of reusing the reasoning answers from its GitHub repository. Similarly, we re-run tests for other models using our RapidAPI accounts. We employ gpt-3.5-turbo-16k and gpt-3.5-turbo-1106 as GPT evaluators, with the results shown in Table 5.

First, TP-LLaMA still outperforms the models without preference learning, further validating the effectiveness of our method. However, the absolute pass rates depend heavily on the specific GPT version. We observe notable differences in preferences and consistency across GPT versions. After repeating the evaluation of each sample 7 times, we find that gpt-3.5-turbo-1106 is more likely to mark a sample as passed, while gpt-3.5-turbo-16k tends to judge it as not passed. This difference mainly stems from how each version assesses whether a query is solvable. Additionally, gpt-3.5-turbo-16k shows greater consistency across the 7 evaluations, meaning it is more likely to produce the same inference repeatedly. This highlights the importance of selecting the appropriate GPT version for evaluation, as relative scores may be more meaningful than absolute ones.

Furthermore, we observe that the gap between TP-LLaMA and ToolLLaMA narrows under the new evaluation. We believe this is due to two factors: (1) The models have different preferences formed during their respective training processes. TP-LLaMA tends to avoid giving up on reasoning and attempts partial answers, whereas ToolLLaMA is more likely to abandon a task entirely, leading to a complete failure. However, this gap narrows due to the use of GPT to evaluate whether the task is solvable. (2) During this supplementary experiment, our RapidAPI accounts have access limits (some APIs even can only be accessed 5 times per month per account), reducing the number of valid samples in the test sets. This particularly affects complex multi-tool reasoning tasks, where TP-LLaMA usually excels, making its performance gains appear smaller.

Additionally, the results we report for ToolLLaMA are still lower than those in Qin et al. [2024], likely due to shifts in the distribution of real-world APIs, which may make certain test samples unsolvable. Moreover, some features ToolLLaMA learned from past environments may not fully

---

[8]https://huggingface.co/ToolBench/ToolLLaMA-2-7b-v2

Table 5: New Pass Rate Experiment Results

| Model | G1-Ins. | G1-Tool | G1-Cat. | G2-Ins. | G2-Cat. | G3-Ins. | Avg |
|---|---|---|---|---|---|---|---|
| **gpt-3.5-turbo-16k** | | | | | | | |
| ToolLLaMA | **0.29** | 0.35 | 0.40 | 0.33 | 0.29 | 0.25 | 0.32 |
| LLaMA with SFT | 0.22 | 0.29 | 0.39 | 0.28 | 0.29 | 0.28 | 0.29 |
| **TP-LLaMA (ours)** | 0.27 | **0.37** | **0.48** | **0.35** | **0.36** | **0.35** | **0.36** |
| **gpt-3.5-turbo-1106** | | | | | | | |
| ToolLLaMA | 0.35 | **0.42** | 0.39 | 0.63 | 0.59 | 0.68 | 0.51 |
| LLaMA with SFT | 0.33 | 0.40 | 0.40 | 0.64 | 0.55 | 0.60 | 0.50 |
| **TP-LLaMA (ours)** | **0.37** | 0.40 | **0.41** | **0.65** | **0.63** | **0.69** | **0.53** |

align with current conditions, resulting in reduced performance. In the future, we can further explore ways to enhance the models performance stability in evolving environments.

## B.2  Case study

We further illustrate the effectiveness of preference learning in improving the tool-usage capabilities of LLMs with a case study focused on the G3-Ins. scenario. To begin, we present the query along with the relevant API documentation

---

### Case Study: Query and Relevant APIs

**Query**: I'm organizing a film festival and I'm looking for award-winning films. Can you search for videos related to "award-winning" on Vimeo? Additionally, fetch the related people in the "film festival" category to invite them as judges. Finally, provide me with a streaming link for a YouTube video with the ID "UxxajLWwzqY".

**Related API Documentation** (parameter information is omitted):

**Name**: getrelatedchannels_for_vimeo
**Description**: Get Related Channels.

**Name**: searchvideos_for_vimeo
**Description**: Search for videos.

**Name**: getrelatedpeople_for_vimeo
**Description**: Get a list of related people for a category.

**Name**: download_stream_for_ytstream_download_youtube_videos
**Description**: Stream or download info.

**Name**: Finish
**Description**: If you believe that you have obtained a result that can answer the task, please call this function to provide the final answer. Alternatively, if you recognize that you are unable to proceed with the task in the current state, call this function to restart. Remember: you must ALWAYS call this function at the end of your attempt, and the only part that will be shown to the user is the final answer, so it should contain sufficient information.

---

During our experiments, the first three APIs above are temporarily inaccessible since they are real-world. So the best response to this query is a YouTube video link with the ID 'UxxajLWwzqY', completing part of the task. We then briefly present the reasoning trajectories from TP-LLaMA and LLaMA with SFT. Note that each reasoning path is limited to at most 4 API calls here.

## Case Study: Reasoning Trajectories from TP-LLaMA

The reasoning tree of TP-LLaMA includes 5 paths and we show the successful path below.

**Step 1** call [`searchvideos_for_vimeo`] with arguments {"format": "json", "query": "award-winning"}, get the response "...The consumer key passed was not valid..."

**Step 2** call [`getrelatedchannels_for_vimeo`] with arguments {"category": "film festival", "format": "json"}, get the response "...The consumer key passed was not valid..."

**Step 3** call [`download_stream_for_ytstream_download_youtube_videos`] with arguments {"is_id": "UxxajLWwzqY"}, get the response "...'id': 'UxxajLWwzqY', 'title': 'Icona Pop - I Love It (feat. Charli XCX) [OFFICIAL VIDEO]', ..."

**Step 4** call [`Finish`] with arguments {"return_type": "give_answer", "final_answer": "The streaming link for the YouTube video with ID UxxajLWwzqY is [link]..."}, get the response "successfully giving the final answer"

## Case Study: Reasoning Trajectories from LLaMA with SFT

The reasoning tree of LLaMA with SFT includes 8 paths. There is no successful path since the model finally gives up. All the paths are listed below (the numbers in brackets represent the unique id of each node).

**Path 1** `searchvideos_for_vimeo` (1) -> `searchvideos_for_vimeo` (2) -> `searchvideos_for_vimeo` (3) -> `getrelatedpeople_for_vimeo` (4)

**Path 2** `searchvideos_for_vimeo` (1) -> `searchvideos_for_vimeo` (2) -> `searchvideos_for_vimeo` (3) -> `getrelatedchannels_for_vimeo` (5)

**Path 3** `searchvideos_for_vimeo` (1) -> `searchvideos_for_vimeo` (2) -> `searchvideos_for_vimeo` (6) -> `getrelatedpeople_for_vimeo` (7)

**Path 4** `searchvideos_for_vimeo` (1) -> `searchvideos_for_vimeo` (2) -> `searchvideos_for_vimeo` (6) -> `searchvideos_for_vimeo` (8)

**Path 5** `searchvideos_for_vimeo` (1) -> `searchvideos_for_vimeo` (9) -> `searchvideos_for_vimeo` (10) -> `getrelatedpeople_for_vimeo` (11)

**Path 6** `searchvideos_for_vimeo` (1) -> `searchvideos_for_vimeo` (9) -> `searchvideos_for_vimeo` (10) -> `searchvideos_for_vimeo` (12)

**Path 7** `searchvideos_for_vimeo` (1) -> `searchvideos_for_vimeo` (9) -> `searchvideos_for_vimeo` (13) -> `Finish with give up and restart` (14)

**Path 8** `searchvideos_for_vimeo` (1) -> `getrelatedchannels_for_vimeo` (15) -> `Finish` (16)

We observe that because LLaMA with SFT repeatedly tries inaccessible APIs (possibly using different arguments) without first accessing the accessible YouTube API, it finally mistakenly chooses to give up reasoning and is unable to give a partial answer. In contrast, TP-LLaMA successfully calls the YouTube API to provide the best possible answer while using fewer inference steps.

## C   Limitations

While this work demonstrates promising results, it also has some limitations. First, the performance of our approach relies on the quality of the decision tree. We parse preference pairs from trajectories that experts naturally explore, though the quality of these trajectories still requires evaluation. Manually introducing suboptimal branches at specific nodes might provide a more effective approach. Additionally, our method currently does not compare preferences between steps on failure paths, suggesting room for improved data utilization. Finally, our approach requires inputting all historical information along the path at each reasoning step, which can be time-consuming. Implementing summary steps during reasoning could help streamline interaction text, assist the model in extracting relevant information, and improve reasoning efficiency.

