# OpenReview forum: "Advancing Tool-Augmented Large Language Models: Integrating Insights from Errors in Inference Trees"
_NeurIPS.cc/2024/Conference — NeurIPS 2024 poster_

### Official Review · Reviewer_jBLW · 2024-07-05

**Soundness:** 3
**Presentation:** 2
**Contribution:** 2
**Rating:** 7
**Confidence:** 4

**Summary:**

This paper proposes a novel framework for improving tool-augmented large language models by incorporating insights from errors in inference trees. It constructs preference data from expert reasoning trajectories and finetune LLMs accordingly. Experiments demonstrate that the finetuned model significantly outperforms baselines in terms of pass rate and win rate, exhibiting better generalization capabilities with unseen APIs.

**Strengths:**

1. The proposed method is straightforward. The construction of a preference dataset makes full use of the failed attempts and can improve data utilization.
2. The ablation experiments are sound. The paper conducts ablation experiments on multiple open source LLMs and the results all demonstrate the superiority of the proposed method.

**Weaknesses:**

1. The paper can benefit from some case studies. Though numbers suggest that TP-LLaMA can master tool-usage instructions better and exhibits stronger generalization capabilities, case studies comparing the reasoning trajectories between TP-LLaMA and LLaMA w/ SFT on different task scenarios is needed to prove the effectiveness of DPO.

**Questions:**

1. See weakness 1, I'd like the authors to provide some case studies on different task scenarios.

2. I'd like the authors to provide a more detailed explanation of their design choices for training hyperparameters:

(a) Could the authors elaborate on how they determine the optimal number of expert trajectories sampled for SFT training (11142 in the paper)?

(b) Similarly, how was the number of preference pairs used for DPO training decided (8202 in the paper)?

To strengthen the analysis, it would be valuable to include ablation studies investigating the impact of these two hyperparameters on model performance.

3. Aside from ToolBench, there are some other benchmarks that can test the tool reasoning capability, e.g. APIBench. Does TP-LLaMA still perform better than the other baselines?

**Limitations:**

None.

---

> ### Author Rebuttal · Authors · 2024-08-06
>
> Many thanks for the constructive reviews! We will revise our paper accordingly.
>
> ------
>
> **Q1**: "The paper can benefit from some case studies. Though numbers suggest that TP-LLaMA can master tool-usage instructions better and exhibits stronger generalization capabilities, case studies comparing the reasoning trajectories between TP-LLaMA and LLaMA w/ SFT on different task scenarios is needed to prove the effectiveness of DPO."
>
> **A1**: Thanks for your suggestion. Here we provide an example of the G3-Ins. scenario. First, we give the query and related API documentation:
>
> + **query**: I'm organizing a film festival and I'm looking for award-winning films. Can you search for videos related to 'award-winning' on Vimeo? Additionally, fetch the related people in the 'film festival' category to invite them as judges. Finally, provide me with a streaming link for a YouTube video with the ID 'UxxajLWwzqY'.
>
> + **related API documentation** (parameter information is omitted):
>
>   + **name:** getrelatedchannels_for_vimeo, **description**: Get Related Channels.
>   + **name:** searchvideos_for_vimeo, **description**: Search for videos.
>   + **name**: getrelatedpeople_for_vimeo, **description**: Get a list of related people for a category.
>   + **name**: download_stream_for_ytstream_download_youtube_videos, **description**: Stream or download info.
>   + **name**: Finish, **description**: ...
>
>   (Ps: during our experiment, the first three APIs are temporarily not accessible.)
>
> Next, we briefly present the reasoning trajectories of TP-LLaMA and LLaMA w/ SFT respectively. Note that we limit a path to contain at most 12 nodes (ie. at most 4 API calls), and a tree contains at most 200 nodes. We require the model to answer as much content of the query as possible.
>
> + The reasoning tree of TP-LLaMA includes 34 nodes and 5 paths. The successful path is:
>
>   1. call [searchvideos_for_vimeo] with arguments {"format": "json", "query": "award-winning"}, get response "...The consumer key passed was not valid...",
>
>   2. call [getrelatedchannels_for_vimeo] with arguments {"category": "film festival", "format": "json"}, get response "...The consumer key passed was not valid...",
>
>   3. call [download_stream_for_ytstream_download_youtube_videos] with arguments {"is_id": "UxxajLWwzqY"}, get response "...'id': 'UxxajLWwzqY', 'title': 'Icona Pop - I Love It (feat. Charli XCX) [OFFICIAL VIDEO]', ...",
>
>   4. call [Finish] with arguments {"return_type": "give_answer", "final_answer": "The streaming link for the YouTube video with ID UxxajLWwzqY is [link]..."}, get response "successfully giving the final answer".
>
>      (Ps: here we use [link] to replace the real URL in the final answer, because external links are not allowed in rebuttal.)
>
> + The reasoning tree of LLaMA w/ SFT includes 49 nodes and 8 paths. There is no successful path. Here are some of the failed paths:
>
>   1. searchvideos_for_vimeo -> searchvideos_for_vimeo -> searchvideos_for_vimeo -> getrelatedpeople_for_vimeo
>
>   2. searchvideos_for_vimeo -> searchvideos_for_vimeo -> searchvideos_for_vimeo -> getrelatedchannels_for_vimeo
>
>   3. searchvideos_for_vimeo -> searchvideos_for_vimeo -> searchvideos_for_vimeo -> Finish with give up and restart
>
>   4. ...... (omit due to space limit)
>
>   Because the model repeatedly tried inaccessible APIs (possibly using different arguments) without accessing [download_stream_for_ytstream_download_youtube_videos] first, it failed to give part of the answer in the end.
>
> Based on this case, we can see that TP-LLaMA can reduce unnecessary attempts of failed paths, explore uncalled APIs, and obtain accessible answers faster. Due to the space limit, we will discuss case studies in other scenarios in the revised version.
>
> ------
>
> **Q2**: "I'd like the authors to provide a more detailed explanation of their design choices for training hyperparameters... To strengthen the analysis, it would be valuable to include ablation studies investigating the impact of these two hyperparameters on model performance."
>
> **A2**: Thanks for the question. We provide an explanation of our design choices regarding training hyperparameters:
>
> + We initially filtered 42,192 reasoning trees with branching nodes from Toolbench, which led to 69,393 DPO samples and 184,816 SFT samples (as each query may correspond to multiple samples).
>
> + After allocating a small part as a validation set, we sampled training sets of different sizes based on these samples. The sampling methods include: by query and by sample. For sampling by query, the SFT training set ranged from 2,500 to 10,000 queries, and the DPO training set ranged from 5,000 to 32,192 queries, yielding nine combinations. For sampling by sample, SFT varied from 10,000 to 183,561, and DPO from 10,000 to 68,951, yielding seven combinations.
> + We tested most of these training sets and observed that increasing the size may lead to decreased model performance in scenarios requiring generalization, such as G3_instruction (e.g., with settings {SFT: 44,412, DPO: 41,226}, the pass rate dropped to 0.36), possibly due to overfitting. Consequently, we selected the set {SFT: 11,142, DPO: 8,202} finally. We will include the ablation studies in the revised version.
>
> ------
>
> **Q3**: "Aside from ToolBench, there are some other benchmarks that can test the tool reasoning capability, e.g. APIBench. Does TP-LLaMA still perform better than the other baselines?"
>
> **A3**: Thanks for the question. Currently, TP-LLaMA relies on relevant API documentation provided for each query in instructions, which is not available in most benchmarks like APIBench. So we are temporarily unable to directly test its performance on other benchmarks. In our future work, we will add an API retriever to replace the pre-annotated relevant APIs, and then test the effect of TP-LLaMA on other benchmarks.
>
> ------

---

> > ### Comment · Reviewer_jBLW · 2024-08-09
> >
> > Good work. My questions have been addressed. I hope the authors can add case studies in the appendix in the revised version. I therefore vote for acceptance of this paper.

---

> > > ### Author Response · Authors · 2024-08-09
> > >
> > > Thank you for your positive feedback and for raising the score. We really appreciate your suggestion and will add the case studies in our revised version as you recommended.

---

### Official Review · Reviewer_9yS8 · 2024-07-09

**Soundness:** 2
**Presentation:** 2
**Contribution:** 2
**Rating:** 5
**Confidence:** 5

**Summary:**

Empowering Large Language Models (LLMs) with external tools has been considered as an important research direction to extend LLMs' task scope. However, during the tool utilization of LLMs, it usually relied on successful tool execution path. Therefore, how to instruct LLMs to employ successful paths is an important element for using tools. To address this point, this paper introduces a ToolPreference dataset that collect successful and failure paths from ToolBench, and then deploy a preference-based fine-tuning based on the proposed ToolPreference dataset. Experimental results demonstrate that the proposed method can achieve better performance in using tools.

**Strengths:**

**Strengths**

The contribution of this paper can be summarized as two parts:
1. This paper introduces a ToolPreference dataset.
2. This paper introduces a fine-tuning method by using DPO algorithm based on the proposed ToolPreference dataset.

**Weaknesses:**

**Weaknesses**
1. The contribution of this paper is limited. Its main contribution is based on DFSDT and then use DPO method for fine-tuning. Compared with ETO, it just replaces the feedback from environments by using ToolPreference.
2. In the proposed ToolPreference dataset, it just uses successful path and failure path as preferred and dispreferred data. However, I think such a selection is too easy as model will never choose any dispreferred data. I think directly training on these preferrence data can also produce better performance.
3. Besides, Tool using also require model to be generalized to other scenarios. Is the proposed model also suitable for other scenarios?
4. The problems in tool usage usually involve multiple aspects, mentioned by ToolQA [a], like parameters incorrection, choose incorrect tools. I think besides to train tool preference, how to enable LLMs to successfully use Tools is more important. And authors can also analyze whether the proposed method can alleviate these errors when using Tools.


**Minor issues**
1. I didn't see any dataset about ToolPreference in supplemental materials. I think if this dataset is considered as a contribution, it should also be reviewed.

[a] A Dataset for LLM Question Answering with External Tools

**Questions:**

1. Can the proposed method be generalized to other benchmarks? It seems that the proposed method is just applied to ToolBench.
2. Did you involve any human evaluation? Pass rate and win rate are just two objective metrics to evaluate whether the model can execute the answer path and whether the solution path is preferred. It cannot guarantee the obtained answer is optimal. Therefore, human evaluation is also required in this task.

**Limitations:**

Please see my above comments.

---

> ### Author Rebuttal · Authors · 2024-08-06
>
> Many thanks for the constructive reviews! We will revise our paper accordingly.
>
> ------
>
> **Q1**: "In the proposed ToolPreference dataset, it just uses successful path and failure path as preferred and dispreferred data. However, I think such a selection is too easy as model will never choose any dispreferred data. I think directly training on these preferrence data can also produce better performance."
>
> **A1**: Thanks for the comment. Below we give an explanation of our approach.
>
> + The "preferred" and "dispreferred" data in ToolPreference actually reflect human preferences based on whether the reasoning path is successful, rather than the model's tool preferences. Because humans always prefer paths that lead to successful reasoning and more correct answers. By constructing the preference dataset this way, we aim to align human and model preferences.
> + Additionally, expert trajectory data for our SFT training comes from the same DFSDT samples as ToolPreference, meaning our SFT training includes all preferred data. However, our results indicate that training only on preferred data leads to poorer performance than utilizing ToolPreference for DPO training.
>
> ------
>
> **Q2**: "Tool using also require model to be generalized to other scenarios. Is the proposed model also suitable for other scenarios?"
>
> **A2**: Thanks for the question. ToolBench contains a wide range of tool usage scenarios across 49 categories, such as news, music, and advertising. Some categories are reserved for testing and do not appear during training. So we have evaluated our model's generalization ability based on ToolBench in our paper, specifically from three perspectives:
>
> 1. Instruction Scenario: Evaluating the model's reasoning for new instructions.
> 2. Tool Scenario: Assessing the model's application of unseen tools within known categories.
> 3. Category Scenario: Evaluating the model's ability to use tools from new categories.
>
> Since different tool categories focus on different task scenarios, the results in our paper, particularly from the Category Scenario, offer valuable insights into our model's generalization performance, which outperformed baseline models.
>
> ------
>
> **Q3**: "The problems in tool usage usually involve multiple aspects, mentioned by ToolQA [a], like parameters incorrection, choose incorrect tools. I think besides to train tool preference, how to enable LLMs to successfully use Tools is more important. And authors can also analyze whether the proposed method can alleviate these errors when using Tools."
>
> **A3**: Thanks for the comment. We provide a brief explanation below.
>
> + Firstly, it is important to clarify that our ToolPreference dataset construction relies on correct tool and parameter selection, as expert trajectories that succeed within a limited path length are more likely to select the right tools and input correct parameters.
> + As explained in **A1**, ToolPreference reflects human preferences rather than a mere model preference toward tools. Our goal in optimizing based on ToolPreference is to enable models to make better choices more rapidly while minimizing unnecessary exploration.
> + Our experimental results indicate that our method effectively reduces errors in tool selection and parameter generation, as our model achieves higher rates with fewer reasoning steps. However, quantitatively analyzing these error reductions is challenging due to the need for extensive human annotation, and we will try to include this in our revised version.
>
> ------
>
> **Q4**: "I didn't see any dataset about ToolPreference in supplemental materials. I think if this dataset is considered as a contribution, it should also be reviewed."
>
> **A4**: Thanks for the suggestion. We will make our dataset public after the paper is accepted.
>
> ------
>
> **Q5**: "Can the proposed method be generalized to other benchmarks? It seems that the proposed method is just applied to ToolBench."
>
> **A5**: Our method can be applied to other benchmarks. Here’s a brief overview of our process:
>
> 1. Based on the DFSDT framework, using GPT to generate annotated reasoning trees for training samples of any benchmark.
> 2. Using the expert annotations from the previous step, create a preference dataset through our proposed method.
> 3. Perform SFT and DPO training based on pre-train models. SFT uses instruction-tuning data from successful paths in the expert annotations, and DPO employs the preference dataset.
>
> The above process is universal and does not depend on Toolbench, so it can be applied to other benchmarks. Furthermore, we plan to integrate an API retriever in future work, allowing TP-LLaMA to be used across benchmarks without additional training.
>
> ------
>
> **Q6**: "Did you involve any human evaluation? Pass rate and win rate are just two objective metrics to evaluate whether the model can execute the answer path and whether the solution path is preferred. It cannot guarantee the obtained answer is optimal."
>
> **A6**: Thanks for your suggestion, we will enhance our evaluation in the revised version accordingly. Due to the variety of APIs in ToolBench, human evaluation is complex and time-consuming. And some queries yield open answers, making it challenging for human annotators to determine the optimal answer. So we didn't involve large human evaluation but used the common method of taking ChatGPT as an expert to evaluate.
>
> In addition, it should be emphasized that the win rate measures both the preference for solution paths and final answers. Specifically, in win rate evaluation, when both our model's answer and expert annotations are not empty, we use ChatGPT to assess whether each answer solves the problem before comparing reasoning paths. Qin et al., 2024 [1] have shown that the win rate correlates well with human preferences, making it a valid metric for our assessment.
>
> ------
>
> [1] Qin, Yujia, et al. Toolllm: Facilitating large language models to master 16000+ real-world APIs. *ArXiv preprint*, arXiv:2307.16789v1.

---

> ### Comment · Reviewer_9yS8 · 2024-08-13
>
> I didn't have any concern about method but still have some questions about dataset construction.
>
> If we want to apply this method other benchmarks, it also requires us to use DFSDT to get a trajectory tree and then find a solution. Such a solution is to find a way to execute API calls not from the perspective of human preference. And please note, ToolBench is also just a instruction dataset not from real-world scenarios or human data. So, it just represents a solution to find executable tools, not human perference. And it seems you did not conduct any human verification on ToolPreference data (line 187 - line 194). Therefore, I think the proposed method can help model to better find a solution to execute tools but not from human preference. As this dataset is not released and I cannot check this dataset, authors also don't provide any human evaluation in the response. So I will remain some concerns about the dataset and whether it can really align human preference. But I can increase score to 5 as I didn't have any issues about method.

---

> > ### Author Response · Authors · 2024-08-13
> >
> > Many thanks for your constructive feedback and support! Currently, given the complexity of multi-tool usage tasks, we adopt the common approach employed by many existing studies, using GPT as a surrogate for human experts to derive ground truth solutions.
> >
> > We fully understand your concerns about the dataset's alignment with human preferences. In our revised version, we plan to try several strategies to address this issue, including introducing a human validation and correction step during the construction of the ToolPreference dataset, and incorporating a human evaluation process for the model's solution paths in our experiments.
> >
> > Thanks again for your thoughtful review!

---

### Official Review · Reviewer_d56d · 2024-07-18

**Soundness:** 3
**Presentation:** 3
**Contribution:** 3
**Rating:** 7
**Confidence:** 4

**Summary:**

The authors aimed to enhance the tool-augmented language model (LLM) by efficiently learning from the failed explorations that previous works had overlooked. Step-wise preference data were extracted by pairing the branch nodes with the corresponding child nodes from the successful path in the tool-using decision tree. A base model was fine-tuned using DPO on this new dataset, in addition to SFT using only successful trajectories, resulting in ToolPrefer-LLaMA. Evaluation metrics included pass rate and win rate, and assessments across six test scenarios demonstrated superior performance compared to baselines, along with a reduction in inference steps. Furthermore, the authors conducted an ablation study on several base models to validate the effectiveness and generalizability of their method.

**Strengths:**

- This paper proposes a method to leverage the inherent preferences embedded in failed trajectories from existing benchmark datasets, without imposing too much computational overhead. It improves data utilization.
- The ablation study demonstrates the efficiency and portability of this method across different base models, indicating its potential for generalizability and immediate applicability to existing systems.
- This method is practical and logically intuitive, making it likely to be adopted in future tool-augmented training processes.

**Weaknesses:**

- Handling of "Error" Nodes in the ToolBench Dataset
The proposed preference pair generation method does not account for "error" nodes in the original ToolBench dataset (see Figure 4. from https://arxiv.org/abs/2307.16789). This oversight could lead to the generation of incorrect preference pairs, such as `0-9-12(api1_failed) > 0-9-10(api1_succeed)`, potentially skewing the results.

- Pass Rate Metric
The authors mentioned in Section 4.1 that the pass rate evaluation does not verify whether the given answer meets the requirements of the instruction. This could result in inflated scores. Additionally, the criteria for filtering out false positives and the details of the filtering process are not specified, which is a significant oversight.

- Evaluation Settings
The paper introduces six test scenarios that closely resemble those described in ToolBench. However, the detailed testing dataset settings are not specified. Clarification is needed on whether these datasets are the same as those used by ToolBench or if they have been resampled.
The specific version of ChatGPT used in the baseline is not mentioned. Knowing the exact version (e.g., gpt-3.5-turbo-0125, gpt-4-0613, gpt-4-0125-preview) is essential for replicating the results and understanding the capabilities of the baseline model.

- ToolLLaMA Results Report
The paper states that ToolLLaMA results were "directly used," but it is unclear how this was done. If the scores were used directly, they do not match those in the referred publication. If the answers were used for score recalculation, the discrepancy in test performance, such as the pass rate for ToolLLaMA in G3-Ins. (0.17) versus I3-Inst. (66.0%), needs to be explained.

- Data benefit or more post-SFT training
The DPO fine-tuned models on the ToolPreference dataset outperform both commercial models and open-source ToolLLaMA, as well as in-house SFT-trained LLaMA. This improvement might stem more from increased computational investment than from the dataset's inherent benefits. To clarify this, experiments should compare DPO fine-tuned SFT-trained models using different datasets: ToolPreference versus successful paths from the ToolBench dataset. This approach would maintain consistent compute resources and DPO methodology, allowing for a direct comparison of how dataset choice impacts performance.

**Questions:**

- Computation Time Clarification: Could you specify the computation time on the mentioned infrastructure?

- Error Node Handling: How do you plan to address the issue of "error" nodes in the preference pair generation method? Could you provide examples or a detailed explanation?

- Evaluation Criteria: Can you clarify the criteria for what is considered a false positive and the details of the filtering process? This would help in understanding the robustness of the evaluation.

- Dataset Settings: Are the test scenarios and dataset settings identical to those used by ToolBench, or were they resampled, and what scale? This information is necessary for reproducibility and context.

- ChatGPT Version Specification: Which specific version of ChatGPT was used in the baseline? This is important for replicating the results.

- ToolLLaMA Results Usage: How were the ToolLLaMA results used in your comparison? Were the scores used directly, or were the answers recalculated? Clarifying this would help in understanding the validity of the comparison.

- Same amount of post-SFT compute on different dataset: Would you consider conducting additional comparative experiments to address this? if not, could you provide specific reasons why you believe this is not a concern?

**Limitations:**

See above

---

> ### Author Rebuttal · Authors · 2024-08-06
>
> Many thanks for the constructive reviews! We will revise our paper accordingly.
>
> ------
>
> **Q1**: "Computation Time Clarification: Could you specify the computation time on the mentioned infrastructure?"
>
> **A1**: Below, we specify the computation time:
>
> + SFT training: 4.6h for 2 epochs.
> + DPO training: 3.2h for 1 epoch.
> + Inference: 3.4s per API call and 48.7s per query. Computation time varies due to task complexity, network conditions, and API service status.
>
> ------
>
> **Q2**: "Error Node Handling: How do you plan to address the issue of "error" nodes in the preference pair generation method? Could you provide examples or a detailed explanation?"
>
> **A2**: Thanks for the valuable insight. On the one hand, most error nodes appear on successful paths because some APIs are temporarily invalid due to their real-time nature, so the model can't get useful responses while generating correct API calls, not necessarily model decision errors. If the APIs become responsive later, the decisions would yield correct results. On the other hand, the error nodes on successful paths from incorrect API calls or parameter errors are relatively few, thus having minimal impact on results.
>
> Nonetheless, we plan to address this issue in the revised version as follows:
>
> + Use GPT to assess the legality of API call names and parameters, as well as evaluate the reasonableness of preference pairs.
> + Filter out error nodes caused by API response failures in the preferred nodes and assess the impact on results.
>
> ------
>
> **Q3**: "Evaluation Criteria: Can you clarify the criteria for what is considered a false positive and the details of the filtering process? This would help in understanding the robustness of the evaluation."
>
> **A3**: Thanks for the question, we will add more details in the revised version. Here’s a brief explanation:
>
> + False positives are instances where "Finish with final answer" is called but the response lacks substantive meaning (e.g., “I'm sorry, but there are no new arrivals on different platforms”). In practice, we collect common features in false positives (e.g., "sorry", "apologize") and filter responses accordingly to exclude them.
>
> + Pass rate assesses if answers are provided and meaningful. While some scores may be inflated, removing false positives limits this impact. Additionally, in win rate evaluation, we compare the quality of our model reasoning answers with expert annotations, using ChatGPT to assess whether each answer solves the problem. Therefore, combining the results of pass rate and win rate can make a reliable evaluation of the model.
>
> We will improve the pass rate evaluation based on your suggestions and include relevant results in the revised manuscript.
>
> ------
>
> **Q4**: "Dataset Settings: Are the test scenarios and dataset settings identical to those used by ToolBench, or were they resampled, and what scale?"
>
> **A4**: Thanks for this suggestion, we will add details in the revised version. Our test datasets come from ToolBench's open source on GitHub. Test scenarios and dataset settings are identical to theirs: G1-Ins., G1-Tool, G1-Cat., G2-Ins., G2-Cat. each contain 200 samples, G3-Ins. has 100 samples.
>
> ------
>
> **Q5**: "ChatGPT Version Specification: The specific version of ChatGPT used in the baseline is not mentioned. Knowing the exact version is essential for replicating the results and understanding the capabilities of the baseline model."
>
> **A5**: We will specify the ChatGPT version in the revised version.
>
> + For evaluation of win rates, we used the gpt-3.5-turbo-16k-0613 version.
> + The pass rate results we reported for ChatGPT in baselines are computed using test answers provided by ToolBench's open source on GitHub.
>
> ------
>
> **Q6**: "ToolLLaMA Results Usage: How were the ToolLLaMA results used in your comparison? Were the scores used directly, or were the answers recalculated? Clarifying this would help in understanding the validity of the comparison."
>
> **A6**: Sorry for our unclear expression, which may have caused difficulties in understanding. We recalculated the scores of ToolLLaMA using test answers provided by ToolBench's open source on GitHub. Due to space limitations, please check A3 to Reviewer z4Vk for more discussions.
>
> ------
>
> **Q7**: "Same amount of post-SFT compute on different dataset: Would you consider conducting additional comparative experiments to address this? if not, could you provide specific reasons why you believe this is not a concern?"
>
> **A7**: Since DPO is a preference optimization method, we can't use only successful paths from the ToolBench for preference training. And because there are no other preference datasets developed based on ToolBench, we can not directly compare how dataset choice impacts the performance, but we will try to construct preference datasets in other ways for comparative experiments in the revised version.
>
> Currently, we can provide a preliminary comparative experiment to show that the performance improvement should not come from increased compute resources. Specifically, we created a new SFT dataset using the same original samples and size as ToolPreference, and performed one epoch SFT training based on the same post-SFT model. The pass rates we observed were lower than those of the DPO fine-tuned model and most even falling below the post-SFT model:
>
> |                | G1-Ins. | G1-Tool | G1-Cat. | G2-Ins. | G2-Cat. | G3-Ins. | Average |
> | -------------- | ------- | ------- | ------- | ------- | ------- | ------- | ------- |
> | LLaMA with SFT | 0.47    | 0.53    | 0.72    | 0.48    | 0.63    | 0.35    | 0.53    |
> | TP-LLaMA       | 0.55    | 0.65    | 0.80    | 0.62    | 0.67    | 0.61    | 0.65    |
> | New Model      | 0.34    | 0.54    | 0.60    | 0.39    | 0.52    | 0.18    | 0.43    |
>
> This may be a result of overfitting. Clearly, simply increasing the amount of post-SFT compute does not enhance the model's learning of the data.
>
> ------

---

> > ### Comment · Reviewer_d56d · 2024-08-08
> >
> > Thank the authors for their detailed responses. The suggested revised content should be sufficient to address most of my questions. Therefore, I have raised the scores in anticipation of reading the updated manuscript.
> >
> > According to A2, there are some "imputable" nodes in certain trajectories. It seems to me that removing them before pooling the preferred nodes could lead to a better preference dataset, as this would eliminate some avoidable errors from the failed paths that may be involved in your preference dataset.
> >
> > It is interesting that the ToolLLaMA results are not reproducible. In this case, I think the writing in this part should clarify your replication efforts and include some explanations, as you gave to z4Vk, for future researchers' reference.
> >
> > Solo re-SFT on the post-SFT model could address this concern to some extent, but not fully. Comparative experiments ideally should employ DPO and roughly the same amount of compute but on different datasets. I am looking forward to your revised section on this issue.

---

> > > ### Author Response · Authors · 2024-08-09
> > >
> > > Thanks so much for your thoughtful feedback and for raising the scores. We will incorporate your insights into the revised manuscript and address the concerns you've mentioned.

---

### Official Review · Reviewer_z4Vk · 2024-07-22

**Soundness:** 3
**Presentation:** 4
**Contribution:** 3
**Rating:** 7
**Confidence:** 3

**Summary:**

This paper proposes a method to collect preference data from decision trees, making use of the failed trajectories which are previously ignored.  Based on ToolBench (https://arxiv.org/abs/2307.16789) and the decision trees provided in the original dataset, a preference dataset, ToolPreference, is built with the proposed method. DPO is used to optimize LLaMA-2-7B on the new dataset, and the resulting model, TP-Llama, achieves state-of-the-art performance on ToolBench.

**Strengths:**

1. Though the idea of the paper is not completely novel, it's the first to apply such ideas on this specific domain.
2. The paper is well written. The results are sound and solid.
3. The paper clearly expresses its own contribution. Yet, it still needs to clarify a few issues regarding the results of baseline methods.
4. The paper is of moderate significance. It provides a new preference dataset for LLM agents.

**Weaknesses:**

1. The paper is related to the works on process reward, e.g., https://arxiv.org/abs/2310.10080, https://arxiv.org/pdf/2312.08935, yet does not discuss about it in the related works section.
2. Two different methods of preference dataset construction are discussed in the paper (path-wise and step-wise), but there is no result supporting that discussion.

**Questions:**

The result of ToolLLaMA reported in this paper is different from the original paper. l feel like it's due to different definitions of "pass rate", but can you give an explanation on this?

**Limitations:**

The paper lacks a thorough discussion on its limitations. For example, it seems that the performance of the method highly depends on the quality of the decision trees, while "ETO" mentioned in the paper doesn't.

---

> ### Author Rebuttal · Authors · 2024-08-06
>
> Many thanks for the constructive reviews! We will revise our paper accordingly.
>
> ------
>
> **Q1**. "The paper is related to the works on process reward, e.g., arxiv: 2310.10080, arxiv: 2312.08935, yet does not discuss about it in the related works section."
>
> **A1**. Thank you for pointing out these important references. Both the two reference works train a process reward model to evaluate each step in the multi-step reasoning task. Our motivation is similar to theirs, but instead of explicitly performing reinforcement learning on the model with process rewards, we construct a preference dataset based on the characteristics of DFSDT to optimize the preference of the model. Their works provide inspiration for us to improve the way we construct preference data in the future. We will cite them and add more discussions in the revised version.
>
> ------
>
> **Q2**: "Two different methods of preference dataset construction are discussed in the paper (path-wise and step-wise), but there is no result supporting that discussion."
>
> **A2**: Thanks for the suggestion. Due to code difficulty and time constraints, we are unable to provide experiment results here, but we will add comparative experiments in the revised version. Below we briefly give some theoretical explanations.
>
> + Our model employs step-wise reasoning via a depth-first search, making only one decision (i.e, API call) per interaction, and each step is based on the previously obtained API feedback. The path-wise method focuses on learning an entire reasoning path, which may cause the model to generate the entire decision sequence at once without receiving synchronous feedback from APIs. We believe this approach is fundamentally unsuitable for our step-wise reasoning paradigm. Using a path-wise preference dataset might negatively impact the model's single-step inference performance.
> + The recent work of Lai et al., 2024 [1], who compared vanilla DPO and step DPO, reveals that step-wise preference optimization benefits long-chain reasoning. We are confident that similar results would be observed in the context of complex tool usage.
>
> ------
>
> **Q3**: "The result of ToolLLaMA reported in this paper is different from the original paper. l feel like it's due to different definitions of "pass rate", but can you give an explanation on this?"
>
> **A3**: Thanks for your question. The ToolLLaMA results we reported indeed differ from the original paper, but this disparity is not due to different definitions of pass rate. Below, we give an explanation.
>
> + During our experimental process, we noticed that the original paper had been updated with a revised definition of pass rate and corresponding results. Given resource constraints, we followed the first version's definition [2], using their provided test answers of ToolLLaMA on GitHub (reproduction_data.zip) to compute pass rates. Moreover, it is important to emphasize that we could not reproduce their reported results with either definition.
> + Actually, several researchers on GitHub have also reported inconsistencies between their results and the original results, without a response from the original authors yet. Potential reasons for these differences may include: (1) differences in ChatGPT APIs, despite we use the same version with them, (2) differences in the evaluation environment configurations.
> + While our reported results do not match those in the original paper, it is important to emphasize that our experimental outcomes are still valid and credible. Our evaluations used a consistent setup for all models, so the relative differences between models are meaningful. Moreover, the second version's pass rate is revised upon the first version, and thus, the first version's pass rate also holds a certain reference value.
>
> We will continue our efforts to contact the original authors to align our evaluation settings. Any updates will be included in the revised version of our paper.
>
> ------
>
> **Q4**: "The paper lacks a thorough discussion on its limitations. For example, it seems that the performance of the method highly depends on the quality of the decision trees, while "ETO" mentioned in the paper doesn't."
>
> **A4**: Thanks for the suggestion. We will include a thorough discussion of limitations in the revised version. As noted, the performance of our method relies on the quality of decision trees. In addition, we can not compare preferences between steps on unsuccessful paths, leaving room for data utilization improvement. In future work, we aim to create a general reward model for better preference data construction and filtering. Additionally, our method requires inputting all historical decision information at each inference step, which is time-consuming and increases the difficulty of understanding. To address this, we plan to add a summary step in the reasoning process to shorten interaction text and help the model extract useful information, improving reasoning efficiency.
>
> ------
>
> [1] Lai, Xin, et al. Step-DPO: Step-wise Preference Optimization for Long-chain Reasoning of LLMs. *ArXiv preprint*, arXiv:2406.18629.
>
> [2] Qin, Yujia, et al. Toolllm: Facilitating large language models to master 16000+ real-world APIs. *ArXiv preprint*, arXiv:2307.16789v1.

---

> > ### Comment · Reviewer_z4Vk · 2024-08-08
> >
> > Thank the authors for the detailed response. It addresses all my concerns. Looking forward to your update on A3.

---

> > > ### Author Response · Authors · 2024-08-09
> > >
> > > Thank you very much for your kind reply! We will improve our paper according to your constructive reviews.

---

### Decision · Program_Chairs · 2024-09-25

**Decision:**

Accept (poster)

**Comment:**

This paper presents a method to enhance tool-augmented large language models (LLMs) by leveraging preference data extracted from both successful and failed trajectories. It introduces the ToolPreference dataset, constructed using preference data from the ToolBench dataset. A base LLM is then fine-tuned using DPO on this new dataset, leading to TP-Llama. Experiments demonstrate superior performance of TP-Llama compared to baselines across various metrics and test scenarios.

The paper is strong in the sense that it proposes an intuitively straightforward technical method for an important problem, and experiments demonstrate that the method is effective. The paper is also clearly written.

Reviewers raised several important questions regarding the discussion of related work, method design, and experiments. The authors provided extensive amounts of clarification and some new results in their rebuttal, resolving most of the key concerns.

I strongly recommend that the authors update the camera-ready based on the author-reviewer discussion, clarifying their technical design and adding new results and case studies.